# Abberant Immunoglobulin G Glycosylation in Rheumatoid Arthritis by LTQ-ESI-MS

**DOI:** 10.3390/ijms21062045

**Published:** 2020-03-17

**Authors:** Zhipeng Su, Qing Xie, Yanping Wang, Yunsen Li

**Affiliations:** Institutes of Biology and Medical Science, Soochow University, 199 Ren-Ai Road, Suzhou 215123, China; 20164252004@stu.suda.edu.cn (Z.S.); qjie@suda.edu.cn (Q.X.)

**Keywords:** glycosylation, IgG, mass spectrometry, rheumatoid arthritis

## Abstract

Aberrant glycosylation has been observed in many autoimmune diseases. For example, aberrant glycosylation of immunoglobulin G (IgG) has been implicated in rheumatoid arthritis (RA) pathogenesis. The aim of this study is to investigate IgG glycosylation and whether there is an association with rheumatoid factor levels in the serum of RA patients. We detected permethylated *N*-glycans of the IgG obtained in serum from 44 RA patients and 30 healthy controls using linear ion-trap electrospray ionization mass spectrometry (LTQ-ESI-MS), a highly sensitive and efficient approach in the detection and identification of N-glycans profiles. IgG N-glycosylation and rheumatoid factor levels were compared in healthy controls and RA patients. Our results suggested that total IgG purified from serum of RA patients shows significantly lower galactosylation (*p* = 0.0012), lower sialylation (*p* < 0.0001) and higher fucosylation (*p* = 0.0063) levels compared with healthy controls. We observed a positive correlation between aberrant N-glycosylation and rheumatoid factor level in the RA patients. In conclusion, we identified aberrant glycosylation of IgG in the serum of RA patients and its association with elevated levels of rheumatoid factor.

## 1. Introduction

Rheumatoid arthritis (RA) is a chronic autoimmune disease characterized by activation of the immune system and secretion of autoantibodies [1]. Immunoglobulin G (IgG) glycosylation implicated in RA has been reported in many reports. On its Fc region, IgG has a single N-linked glycan, which is primarily biantennary and present on asparagine 297 (Asn 297), on each heavy chain [2,3]. Changes of the N-linked Fc glycan on Asn 297 have been reported to affect the structural stability and functional activity of IgG, subsequently influencing the immune response [4]. Aberrant N-glycosylation of IgG has been observed in autoantibody-driven diseases, especially in RA [5,6,7], and the pathogenicity of autoantibodies is essentially influenced by their glycosylation profile [8]. The generation of aberrant forms of oligosaccharide structures with a single sialic acid molecule converts an inflammatory IgG into an anti-inflammatory mediator [9], and the generation of unusual structures of the IgG Fc portion with a core fucose residue decreases antibody-dependent cell cytotoxicity (ADCC) via hindering its binding to the FcγRIIIα receptor [10]. Thus, determining whether IgG glycosylation is associated with a clinical outcome is significant for understanding the pathogenesis of autoimmune diseases such as RA.

In recent years, IgG or other antibody glycosylation level in the human serum has been ever clearer by various approaches. In normal human serum, although there are regularly different subclasses on immunoglobulin G, the total IgG glycosylation is generally quite constant [11]. Furthermore, different glycosylation patterns of total IgG have also been observed in patients with a number of auto-immune diseases when compared with healthy controls, including rheumatoid arthritis [5,6], systemic lupus erythematosus [12], inflammatory bowel disease [13], primary Sjögren’s syndrome, ankylosing spondylitis, psoriatic arthritis [14], and multiple sclerosis [15]. In patients with all these diseases, N-glycans of serum IgG are missing terminal galactose (IgG G0) when compared with healthy controls.

The identification of N-glycoforms has been made possible by rapid and reproducible glycomic analyses. The field of mass spectrometry (MS) has developed useful tools to detect and identify specific glycoforms and further provide fragmentation data. Protein-bound N-glycans can be released from biological samples such as serum/plasma using peptide-N-glycosidase F (PNGase F). The free glycans can then be analyzed by electrospray ionization (ESI) coupled with online liquid chromatography and matrix-assisted laser desorption/ionization.

It has been reported that rheumatoid factor (RF), a part of the RA classification criteria, binds to IgG independent of the level of IgG galactosylation [16] and that galactosylation and sialysylation of IgG-RF are dramatically lower in RA [17]. In the present study, we obtained comprehensive IgG N-glycan profiling in large cohorts of RA patients and healthy controls by LTQ-ESI-MS and identified all N-glycan structures using multistage MS (MS^n^). In addition, we evaluated whether the altered glycosylation is related to the RF level in serum of antibody-mediated RA. We observed decreased galactosylation and enhanced fucosylation in RA patients compared with healthy controls. Furthermore, aberrant IgG glycosylation levels correlated significantly with RF avidity.

## 2. Results

### 2.1. Comprehensive Profiling of IgG N-Glycans by ESI-MS

In our initial studies, we investigated IgG glycosylation in 44 RA patients and 30 healthy controls, using a recently developed high-throughput method, linear ion-trap electrospray ionisation mass spectrometry (LTQ-ESI-MS), to obtain a comprehensive glycosylation profile from complex biological samples. The purity of IgG purified from serum was assessed by sodium dodecyl sulphate–polyacrylamide gel electrophoresis (SDS-PAGE) (Figure 1). The heavy chain and light chain of IgG were separated from serum of RA patients and healthy controls. In the IgG separation, the largest amount of IgG was fractionated in serum with few proteins (Figure 1). From the two sample sets, 21 potential N-glycan ions, according to our previous paper [18], were detected in the MS^1^ profile spectrum of the permethylated fraction. The representative profiles of permethylated IgG N-glycans derived from healthy controls and RA patients were recorded in the *m*/*z* range of 800–1500 (Figure 2). The relevant composition (under the number) and structure (above the number) of ions identified by sequential MS^n^ were also illustrated. The N-glycan structures are presented in Table 1.

To further identify the structure of the N-glycans, we performed sequential MS^n^ to analyze the fragmental information and the respective chemical structure. With this method, we analyzed the ion at *m*/*z* 937^2+^, corresponding to N4H3F1. The MS^n^ of FucHex3HexNAc4 at *m*/*z* 937^2+^ followed the MS^n^ pathway of *m*/*z* 937^2+^→808^2+^ (1592^+^)→678^2+^ (1333^+^)→576^2+^ (1129^+^)→662^+^ (Figure 3). The loss of ion fragment and the chemical structure are also illustrated. From the MS^2^ spectrum (Figure 3A), the ion at *m*/*z* 808^2+^ (single changed ion: 1592^+^) was the typical fragment which lost a terminal N-acetylglucosamine (GlcNAc; the neutral loss mass 259). The ion at 808^2+^ lost another terminal GlcNAc, and the consequent ion was *m*/*z* 678^2+^ (single changed ion: 1333^+^) with sequential MS^3^ analysis (Figure 3B). With loss of the internal Hex (the neutral loss mass 204), the double charged ion at *m*/*z* 576^2+^ (single changed ion: 1129^+^) was the product ion in the MS^4^ spectra (Figure 3C). As shown in Figure 3D, with loss of fucosylated GlcNAc (GlcNAc-Fuc, the neutral loss mass 467), the ion at *m*/*z* 662^+^ (Man-Man-GlcNAcNa^+^) was left. The major ion fragments in MS^n^ are shown in Table 2.

### 2.2. Aberrant IgG Glycosylation in RA Patients

To further investigate the differentiation of IgG N-glycans between RA patients and healthy controls, we compared the level of N-glycans between the two sample sets. The IgG glycosylation features, which include galactosylation, sialylation, fucosylation, and bisection, were analyzed in the serum of patients (Figure 4). The level of these glycosylation features was calculated as described in Table 3. Total IgG purified from the serum of RA patients shows significantly lower galactosylation (*p* = 0.0012), lower sialylation (*p* < 0.0001), and higher fucosylation (*p* = 0.0063) levels compared with healthy controls. Compared with healthy controls, there was no significant difference in the levels of total bisection in patients with RA (*p* = 0.0712).

### 2.3. Association between Aberrant IgG Glycosylation and RF Rheumatoid Factor Titer in RA

To investigate whether aberrant IgG glycosylation correlates with clinical outcome, we measured RF levels and IgG Fc glycosylation. Based on the titer of RF in the serum, all patients (*n* = 44) were classified into low (*n* = 17, 31.9–96.4 IU/mL) or high (*n* = 27, 105.0–7120.0 IU/mL) subgroups. Compared with healthy controls and with patients with low RF levels, the subgroup of patients with high RF levels exhibited reduced sialylation levels (Figure 5B). In agreement, higher IgG fucosylation was also observed in the subgroup of patients with high RF levels compared with healthy controls and with patients with low RF levels (Figure 5C). Although there was no significant difference in the level of total bisection in patients with RA, the bisection level was decreased in the patients with high RF levels compared with healthy controls and patients with low RF levels (Figure 5D).

Agalactosylated IgG is a set of immunoglobulin glycoforms of the biantennary oligosaccharide terminating in GlcNAc rather than galactose [5]. Figure 1 shows agalactosylated (G0, no galactose), monogalactosylated (G1, one terminal galactose residue) and fully galactosylated (G2, two galactoses) forms of Fc-associated oligosaccharide chains attached to Asp 297 IgG. We analysed the ratios of G1F/G0F (*m*/*z* 1039/937), G2F/G0F (*m*/*z* 1142/937), G1F0/G0F0 (*m*/*z* 953/850) and G2F0/G0F0 (*m*/*z* 1055/850) between RA patients and healthy controls (Figure 5). Significantly decreased galactosylation of IgG was observed in the group of patients with high RF levels compared with patients with low RF levels (*p* = 0.0025) and with healthy controls (*p* < 0.0001) (Figure 5A). Accordingly, the ratios of G1F/G0F (*m*/*z* 1039/937), G2F/G0F (*m*/*z* 1142/937), G1F0/G0F0 (*m*/*z* 953/850) and G2F0/G0F0 (*m*/*z* 1055/850) were reduced in the serum of all patients with RA compared with healthy controls (Figure 5E-H). In the two subgroups of patients, the ratios of G1F/G0F (*m*/*z* 1039/937) and G2F/G0F (*m*/*z* 1142/937) exhibited dramatic difference, but not the ratios of G1F0/G0F0 (*m*/*z* 953/850) and G2F0/G0F0 (*m*/*z* 1055/850). These results confirmed the aberrant IgG galactosylation in RA patients and demonstrated a remarkable change in the ratio of agalactosylation/monogalactosylation and agalactosylation/galactosylation with two different sets of N-glycans in patients.

## 3. Discussion

Here, the N-glycosylation features of IgG and their association with the level of RF were analyzed. LTQ-ESI-MS was used to confirm previous studies documenting aberrant IgG glycans in RA. In this present study, we identified IgG N-glycan profiles in the serum of patients with RA and found remarkably different profiles of glycoforms compared with healthy controls. A high level of IgG that lacks terminal galactose (IgG G0) was observed in patients with RA, which is in line with previous studies reported for other immune diseases [13,19,20]. Our results also indicated significantly lower sialylation, reduced bisection, and increased fucosylation in patients with RA compared with healthy controls. RF, a class of immunoglobulins, has been reported to correlate with elevated inflammatory cytokines and a more rapid progression to RA [21,22]. Although dramatically lower galactosylation and sialysylation of IgG-RF have been observed in RA [17], and IgG glycosylation profiles influenced the RF level in an experimental arthritis model [23], the association between IgG glycosylation and RF titers has been reported rarely. Access to a substantial collection of RA subjects with different RF levels enabled us to observe the association of IgG glycosylation with RF avidity in RA. In this study we found a positive correlation between the IgG glycosylation and RF level, similar to previous correlation between aberrant IgG galactosylation and the disease activity in a gender specific manner. However, very limited, we did not obtained every clinical index, with the exception of the rheumatoid factor titer.

The conformation of the Asn 297 biantennary oligosaccharide has been shown to influence the functional activity of the IgG molecule via C1q of the complement system and Fc gamma receptors (FcγR) on immune cells [24]. IgG deficient in terminal galactose (IgG G0) may combine with autoantibodies, forming an immune complex that deposits in synovial tissue; this process is directly implicated in the pathogenicity of RA [25,26,27]. Furthermore, the addition of a terminal sialic acid to the Fc region by 2,6-sialyltransferase promoted anti-inflammatory activity [9,28,29,30]. More importantly, deficiencies in galactose will also impact sialylation, due to the need for galactose for the addition of terminal sialic acids [3]. Although engineered IgG sialylation could convert inflammatory IgGs into anti-inflammatory mediators in animal model [30], more work is needed to see if this is the causative agent for the induction of autoimmune reaction. Consistent with previous studies, deceased galactosylation and sialylation of total IgG was observed in patients with RA in this present study. Furthermore, our obtained results also indicated that changes of galactosylation and sialylation positively associated with RF titer.

As in earlier studies, the level of sG0/G1 and G0/G2 in the serum has a significant correlation with disease activity in RA and inflammatory bowel disease (IBD) patients, indicating that G0/G1 and G0/G2 levels could be useful serum glycobiomarkers for the diagnosis of RA. To further explore the correlation between IgG galactosylation content and disease activity, we compared under-galactosylation ratios with the RF titer. In this study, the serum levels of G1F/G0F and G2F/G0F obviously (*p* < 0.05) reduced in RA patients with high RF level, while the levels of G1/G0 and G2/G0, which lack of a core fucose in the Fc fragment, were not found significant differences in RA patients with high and low RF level. These results further suggested that G1F/G0F and G2F/G0F levels may be promising serum biomarkers for RA disease activity and clinical diagnosis.

Recently, bisecting GlcNAc has been recognized as a suppressor of terminal modifications of N-glycan, which may further influence biological functions of glycans on glycoproteins [31]. However, the effects of bisecting GlcNAc on functional properties of IgG are not well understood. As reported in a previous study, 18% of IgG glycans contain bisecting GlcNAc and they could significantly influence the structure of the glycan on IgG [32]. Glycosylation patterns of total IgG, which carry a bisecting N-acetylglucosamine, have been observed to temporarily decrease in certain conditions, including rheumatoid arthritis during pregnancy and anti-neutrophil cytoplasmic antibodies (ANCA)- associated vasculitis (AAV) [33,34,35]. Interestingly, although the decrease in bisecting GlcNAc on IgG glycans did not reach statistical significance in all patients with RA, which is consistent with the previous data [36], bisection level in patients with a high RF titer was lower when compared with patients with low RF titer in our obtained results.

More recent works have revealed that fucosylation plays an important role in influencing the functional activity and the structure of IgG. The absence of fucose residues directly boosts ADCC activity by dramatically enhancing binding to FcγRIIIa [37,38]. Antibodies lacking the core fucose show an increased autoantibody affinity by carbohydrate–carbohydrate interactions between glycans of the receptor and the Fc fragment [39]. Specifically, fucosylation weakens the stabilization between the antibody and FcgRIIIa’s glycan [40]. As previously reported, several specific antibodies display a significantly increased levels of fucosylated glycans in patients [19,35,41]. In the current study analyzing serum total IgG we similarly observed an increase of core fucosylation in RA patients, and in particular with high RF titer, which is thought to enhance the pro-inflammatory activity of IgG molecules. What is different from prior findings is that the general degree of core fucosylation is lower in our present study, approximately in the range of 40–60%, which yet is in line with observations for N-glycans in colorectal cancer tissues in our previous work [42]. These differences may be largely caused by the N-glycan analysis method using mass spectrometry. We analyzed more various N-glycans on IgG increasing the sum of the intensity of total N-glycan peaks, which may contribute to the relatively low general degree of glycosylation in our study.

The mechanisms governing the IgG glycosylation to autoimmune diseases are increasingly attractive for therapeutic application [43,44,45]. Increasingly, B cells, plasma cells, cytokine milieu, and extracellular glycosytransferases have been implicated in altering IgG Fc glycoforms in autoantibody-driven inflammation [7,43,46,47,48,49]. The sialylation could be regulated by sialyltransferases in interleukin-23–activated T helper 17 cells and further converts arthritogenic IgG into anti-inflammatory mediators in experimental arthritis [7,49]. Although the exact cause of the lower galactose levels on IgG in whole serum of RA patients remains indistinct, reduced galactosyltransferase (GalT) activity has been reported in B cells from RA patients and increased GalT content also was reported in synovial tissue and inflammatory cells of RA patients [50,51]. With growing evidence for monitoring the IgG glycosylation, additional glycoengineering of IgG antibodies using endoglycosidase-based glycosynthases could improve the safety, functionality, and efficacy of these antibodies [52].

These findings may help to explain the association between RF level and aberrant N-glycosylation, and have important implications for aberrant N-glycosylation in the development of clinical diagnosis.

## 4. Materials and Methods

### 4.1. Patients and Controls

Serum samples were obtained from 44 patients with RA and 30 healthy controls at Suzhou Hospital of Traditional Chinese Medicine (Suzhou, China) and Huaian Second People’s Hospital (Huaian, China). All experiments were approved by the Medical Ethics Committee of Soochow University. Informed consent was obtained from each participant. Patient characteristics are summarized in Table 4. Serum aliquots were frozen immediately after collection and stored at −80 °C

### 4.2. Preparation of Total Immunoglobulin and IgG

First, total immunoglobulin was obtained in the serum of RA patients and healthy controls; then IgG was purified based on a previously reported method [53]. Briefly, serum (500 μL) separated from each sample by centrifugation was diluted with phosphate-buffered saline (PBS) (1 mL, pH 7.4), and then 1.5 mL of saturated ammonium sulfate solution was added dropwise and mixed by vortex. The achromatous emulsion was centrifuged, and the deposit was dissolved in PBS (200 μL, pH 7.4) as total immunoglobulin. The collected immunoglobulin was used for IgG purification. The IgG was collected by using protein G columns (GenScript, Nanjing, China). The concentration of proteins was determined using the bicinchoninic acid assay kit (Beyotime biotechnology, Nanjing, China), and the purity was identified by SDS-PAGE.

### 4.3. Release and Solid-Phase Extraction of N-Glycans

A 500-μg aliquot of freeze-dried IgG was dissolved in 50 μL of 1× denaturing buffer (New England Biolabs, Ipswich, MA, USA) in water (water used in all experiments was filtered with the Milli-Q filtration system). The IgG was fully denatured with boiling water, and then 10× NP-40 buffer (10 μL), 10× G7 buffer (10 μL), and 2.5 μL of PNGase F (New England Biolabs) were mixed and incubated at 37 °C for 16 h. Glycans were purified using Sep-Pak C18 cartridges (Waters, Beijing, China) according to the method reported in our previous paper [18]. The collected samples were dried under a vacuum for permethylation.

### 4.4. Permethylation of N-Glycans

N-glycans obtained from glycoproteins were permethylated [54]. Briefly, sodium hydroxide (40–60 mg) and iodomethane (100 µL) were added to the collected N-glycans dissolved in dimethyl sulphoxide (150 µL; Tedia, Fairfield, OH, USA) and mixed by shaking vigorously for 1 h at room temperature. The permethylation reaction was stopped with the addition of 2 mL of water. Dichloromethane (2 mL) was added and mixed vigorously. The permethylated N-glycans were extracted with dichloromethane and rinsed repeatedly (10 times) with 2 mL of water. Following the final wash, the aqueous phases were discarded, and the dichloromethane phase was transferred to a clean tube and dried under vacuum.

### 4.5. Analysis of N-Glycans Using Linear Ion-Trap Electrospray Ionisation Mass Spectrometry (LTQ-ESI-MS)

After permethylation, structural analysis of N-glycans was performed in a linear ion-trap mass spectrometer (Thermo Finnigan, San Jose, CA, USA) in positive ion mode. Glycans dissolved in methanol were loaded at 0.5 µL/min at a sheath gas flow rate of 2 arb, a spray voltage of 3.50 kV, a capillary voltage of 35 V, a capillary temperature of 230 °C, a tube lens of 120 V, an injection time of 100 ms, an activation time of 30 ms, an activation Q-value of 0.250, and an isolation width of *m*/*z* 1.5. To further identify the structure of N-glycans, MS^n^ spectra were obtained. All samples were analyzed as sodium-adducted positive ions.

### 4.6. Statistical Analysis

The sample was introduced manually with the same injection time (100 ms) and the same volume (2 μL). With the same retention time (1–3 min), the mass spectrum of each sample was extracted after MS profiling of each experimental sample obtained by Xcalibur software version 2.2 (Thermo Fisher Scientific, San Jose, CA, USA). At the time point of strongest ionic strength, the mass charge ratio (*m*/*z*) and the ion intensity of mass peaks were exported by Xcalibur Qual Browser (Thermo Fisher Scientific, San Jose, CA, USA). Relative quantification was conducted with the formula (the intensity of specific N-glycan peak)/(the intensity of total N-glycan peaks) [18,42,55], which normalizes the absolute ion intensities to the total subclass specific N-glycan abundances.

The level of galactosylation was calculated according to the formula: N4H4 + N4H5F1 + N5H5 + N4H5S1 + N5H5F1 + N4H5F1S1 + N4H5F2S1 + N4H5S2 + N5H5F1S1 + 0.5 * (N4H4 + N4H4F1 + N5H4 +N4H4F2 + N4H4S1 + N5H4F1). The level of sialylation was defined by summation of all sialic acids per antenna on N-glycans and calculated by the formula: N4H5S2 + 0.5 * (N4H4S1 + N4H5S1 + N4H5F1S1 + N4H5F2S1 + N5H5F1S1 + N5H6S1). The prevalence of core fucosylation was measured by summing the N-glycan glycoforms which carries one core fucose (N4H3F1 + N4H4F1 + N5H3F1 + N4H5F1 + N5H4F1 + N5H5F1 + N4H5F1S1 + N5H5F1S1 + N4H4F2 + N4H5F2S1). The incidence of bisecting was evaluated by summing all N-glycoforms which carry a bisecting N-acetylglucosamine (N5H3 + N5H3F1 + N5H4 + N5H4F1 + N5H5 + N5H5F1 + N5H5F1S1 + N5H6S1).

Statistical analyses were performed using GraphPad Prism software (GraphPad Software, La Jolla, CA, USA). The presence or absence of statistical differences between controls and patients was evaluated using two-tailed Student *t*-test. *p* < 0.05 indicated statistical significance.

## Figures and Tables

**Figure 1 ijms-21-02045-f001:**
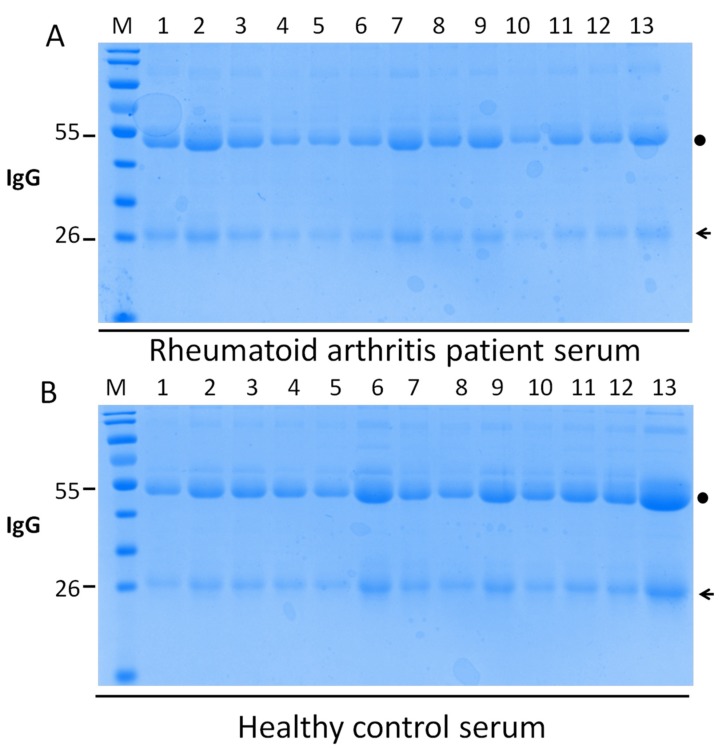
Purity assessment of immunoglobulin G (IgG) from human serum. IgG was purified by protein G affinity chromatography in the serum of rheumatoid arthritis patients (**A**) and healthy controls (**B**), and the heavy chain (black dot) and light chain (arrow) of IgG were separated. Sodium dodecyl sulphate–polyacrylamide gel electrophoresis was performed to determine the purity.

**Figure 2 ijms-21-02045-f002:**
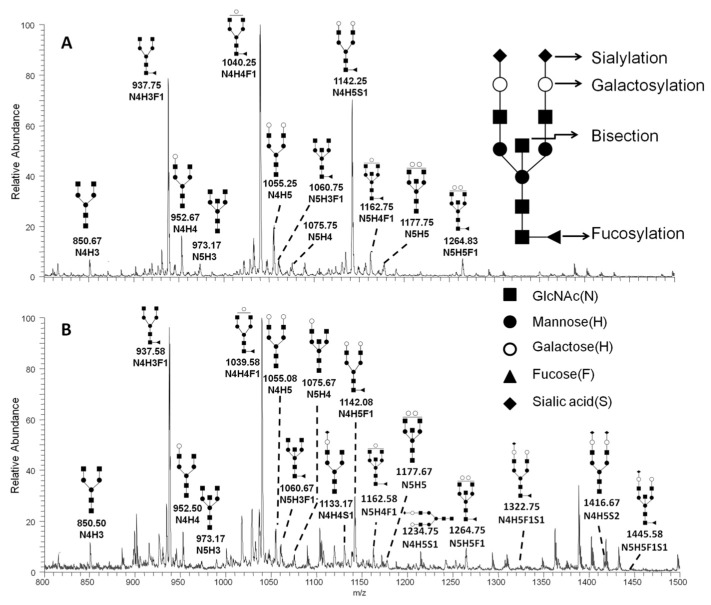
Representative MS^1^ spectrums of N-glycans from healthy control Immunoglobulin G (IgG) (**A**) and rheumatoid arthritis patient IgG (**B**). The representative structure of IgG Fc N-glycan is shown. Based on the core structure (GlcNAc2-Man3-GlcNAc2), galactoses (H, lighter circle), Mannose (H, solid circle), bisecting GlcNAc (N, diamond), sialic acids (S, rhombus), or fucose (F, triangle) can be attached.

**Figure 3 ijms-21-02045-f003:**
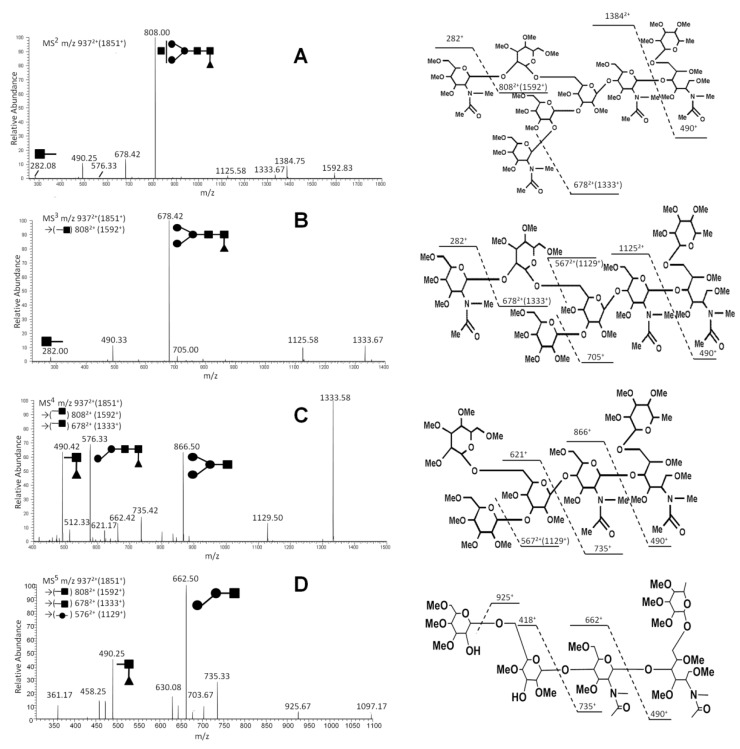
Multistage MS (MS^n^) pathway of the doubly changed ion at *m*/*z* 937 in serum from a rheumatoid arthritis patient. The pathway consists of MS^2^
*m*/*z* 937^2+^ (**A**), MS^3^
*m*/*z* 808^2+^ (**B**), MS^4^
*m*/*z* 678^2+^ (**C**) and MS^5^
*m*/*z* 567^2+^ (**D**), along with their respective chemical structures.

**Figure 4 ijms-21-02045-f004:**
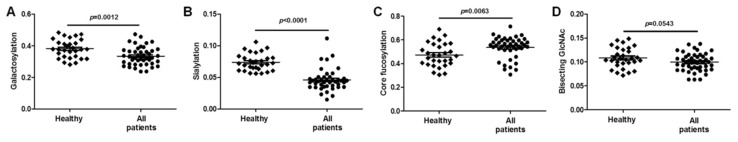
Aberrance of immunoglobulin G (IgG) glycosylation in serum of rheumatoid arthritis (RA) patients compared with healthy controls. IgG glycosylation in serum of patients with RA is different compared with healthy controls. Changes were also observed between patients with three various rheumatoid factor (RF) levels and healthy controls for total galactosylation (**A**), sialylation (**B**), core fucosylation (**C**), and bisecting GlcNAc (**D**). A two-tailed Student’s *t*-test comparison was performed between the healthy controls and patients, and patients with three RF levels. Error bar indicates standard deviation (ns, no significance; *p* < 0.05 indicated statistical significance).

**Figure 5 ijms-21-02045-f005:**
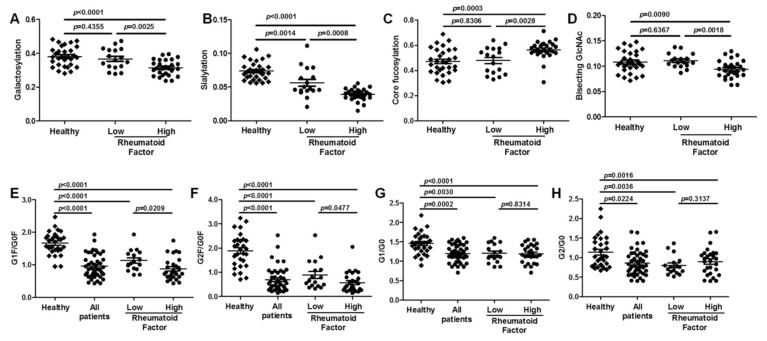
Association between aberrant immunoglobulin G (IgG) glycosylation and rheumatoid factor (RF) titer in rheumatoid arthritis. The IgG galactosylation (**A**), sialylation (**B**), core fucosylation (**C**), bisection (**D**), G1F/G0F (**E**), G2F/G0F (**F**), G1F0/G0F0 (**G**), and G2F0/G0F0 (**H**) for all patients, as well as patients with different RF levels, are aberrant compared with healthy controls. Statistical significance was evaluated using two-tailed Student’s *t*-test. Error bar indicates standard deviation (ns, no significance; *p* < 0.05 indicated statistical significance).

**Table 1 ijms-21-02045-t001:** Major N-glycans in sera of healthy controls and rheumatoid arthritis patients.

*m*/*z*	Charge	Composition	Main Structure	MS^n^ Pathway
850	2+	N4H3	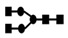	850^2+^→721^2+^ (loss of 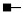 )→591^2+^ (loss of 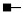 )→866^+^ (loss of 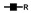 )→662^+^ (loss of 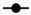 )
937	2+	N4H3F1	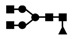	937^2+^→1592^+^ (loss of 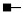 )→ 1333^+^ (loss of 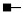 )→1129^+^ (loss of 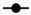 )→ 662^+^ (loss of 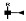 )
953	2+	N4H4	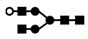	953^2+^→823^2+^ (loss of 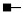 )→ 1159^+^ (loss of 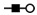 )→866^+^ (loss of 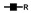 )→ 662^+^ (loss of 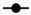 )
973	2+	N5H3	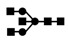	973^2+^→843^2+^ (loss of 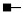 )→ 714^2+^ (loss of 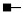 )→584^2+^ (loss of 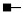 )→ 852^+^ (loss of 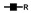 )
1039	2+	N4H4F1	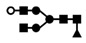	1039^2+^→910^2+^ (loss of 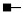 )→ 1333^+^ (loss of 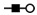 )→866^+^ (loss of 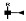 )
1055	2+	N4H4	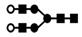	1055^2+^→1623^+^ (loss of 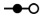 )→ 1329^+^ (loss of 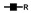 )→866^+^ (loss of 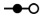 ) →662^+^ (loss of 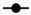 )
1060	2+	N5H3F1	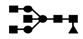	1060^2+^→931^2+^ (loss of 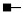 )→ 801^2+^ (loss of 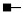 )→672^2+^ (loss of 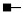 )→852^+^ (loss of 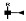 )→648^+^ (loss of 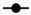 )→444^+^ (loss of 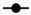 )
1075	2+	N5H4	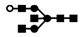	1075^2+^→946^2+^ (loss of 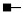 )→ 816^2+^ (loss of 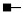 )→1146^+^ (loss of 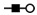 )→853^+^ (loss of 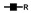 )→648^+^ (loss of 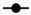 )→444^+^ (loss of 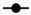 )
1127	2+	N4H4F2	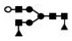	1127^2+^→1768^+^ (loss of 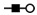 )→ 1301^+^ (loss of 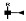 )→867^+^ (loss of 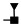 )
1133	2+	N4H4S1	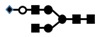	1133^2+^→946^2+^ (loss of 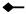 )→ 1419^+^ (loss of 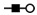 )→1159^+^ (loss of 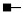 )→866^+^ (loss of 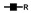 )→662^+^ (loss of 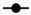 )
1142	2+	N4H5F1	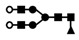	1142^2+^→1797^+^ (loss of 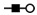 )→ 1330^+^ (loss of 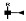 )→867^+^ (loss of 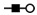 )→ 662^+^ (loss of 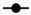 )
1162	2+	N5H4F1	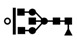	1162^2+^→1033^2+^ (loss of 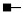 )→ 903^2+^ (loss of 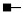 )→1320^+^ (loss of 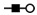 )→852^+^ (loss of 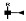 )→648^+^ (loss of 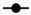 )→444^+^ (loss of 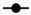 )
1177	2+	N5H5	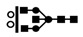	1177^2+^→946^2+^ (loss of 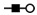 )→ 816^2+^ (loss of 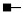 )→1146^+^ (loss of 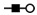 )→852^+^ (loss of 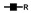 )→648^+^ (loss of 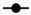 )→444^+^ (loss of 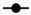 )
1235	2+	N4H5S1	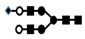	1235^2+^→1047^2+^ (loss of 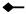 )→ 823^2+^ (loss of 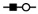 )→1329^+^ (loss of 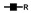 )→867^+^ (loss of 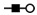 )→ 662^+^ (loss of 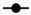 )
1264	2+	N5H5F1	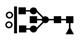	1264^2+^→1135^2+^ (loss of 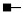 )→ 903^2+^ (loss of 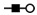 )→1319^+^ (loss of 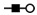 )→852^+^ (loss of 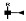 )→ 662^+^ (loss of 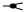 )→458^+^ (loss of 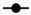 )
1320	2+	N2H10	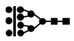	1320^2+^→1173^2+^ (loss of 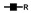 )→ 1696^+^ (loss of 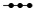 )→1492^+^ (loss of 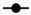 )→1274^+^ (loss of 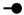 )
1322	2+	N4H5F1S1	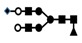	1322^2+^→1135^2+^ (loss of 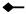 )→ 910^2+^ (loss of 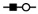 )→1333^+^ (loss of 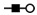 )→866^+^ (loss of 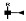 )→662^+^ (loss of 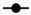 )
1409	2+	N4H5F2S1	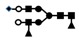	1409^2+^→1221^2+^ (loss of 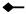 )→ 996^2+^ (loss of 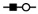 )→1333^+^ (loss of 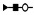 )
1416	2+	N4H5S2	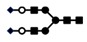	1416^2+^→1228^2+^ (loss of 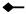 )→ 1041^2+^ (loss of 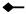 )→1609^+^ (loss of 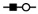 )→1315^+^ (loss of 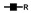 )→866^+^ (loss of 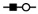 )→662^+^ (loss of 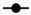 )
1445	2+	N5H5F1S1	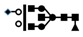	1445^2+^→1257^2+^ (loss of 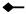 )→ 1033^2+^ (loss of 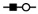 )→903^2+^ (loss of 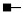 )→1316^+^ (loss of 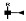 )→852^+^ (loss of 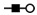 )
1460	2+	N5H6S1	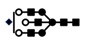	1460^2+^→1272^2+^ (loss of 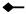 )→ 1040^2+^ (loss of 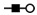 )→894^2+^ (loss of 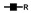 )→669^2+^(loss of 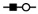 )→852^+^ (loss of 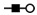 )

**Table 2 ijms-21-02045-t002:** Major ion fragments in MS^n^.

Fragment	Neutral Loss	[M + Na]^+^	Symbol
Terminal Hex	218	241	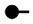
Internal Hex	204	227	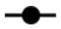
Branched Hex	190	213	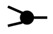
Terminal HexNAc	259	282	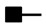
Internal HexNAc	245	268	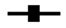
Terminal NeuAc	375	398	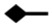
Terminal Fuc	188	211	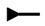
Reducing-end GlcNAc	293	316	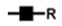
GlcNAc-Fuc	467	490	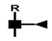
Terminal Hex-HexNac	463	486	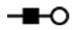
Internal Hex-HexNac	449	472	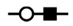
Core Man-GlcNAc	-	444	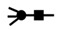

**Table 3 ijms-21-02045-t003:** A review of the description for the N-glycosylation feature.

N-Glycosylation	Description	*m*/*z* of N-Glycans	
Galactosylation	galactoses per antenna on *N*-glycans	monogalactosylated	952, 1039, 1075, 1127, 1133, 1162
		Fully galactosylated	1054, 1142, 1177, 1235, 1264, 1322, 1409, 1416, 1445
Sialylation	*N*-acetylneuraminic (sialic) acids per antenna on *N*-glycans	1133, 1235, 1322, 1409, 1416, 1445, 1460	
Fucosylation	*N*-glycans which carry a core fucose	937, 1039, 1142, 1060, 1162, 1264, 1322, 1445, 1127, 1409	
Bisection	*N*-glycans which carry a bisecting *N*-acetylglucosamine	973, 1060, 1075, 1162, 1177, 1264, 1445, 1460	

**Table 4 ijms-21-02045-t004:** Clinical characteristics of the participant serum samples used in this study.

Characteristics	Patient	Normal
Number ^a^	44	30
Female (%) ^b^	66.67	76.67
Age (year) (median, range)	61 (30–87)	36 (21–44)
RF positivity (%) ^c^	100	-
RF (median, U/mL) ^d^	759.75	-

^a^ For each participant, serum, and peripheral blood were obtained; ^b^ Frequency of female; ^c^ Rheumatoid factor positivity; ^d^ The mean titer of rheumatoid factor.

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
