# Peer review of "Abberant Immunoglobulin G Glycosylation in Rheumatoid Arthritis by LTQ-ESI-MS"

_ijms, 2020, doi:10.3390/ijms21062045_

Round 1

Reviewer 1 Report

None

Reviewer 2 Report

The titled work: Abberant immunoglobulin G glycosylation in rheumatoid arthritis by LTQ-ESI-MS is not novel or new to the LC_MS analysis of IG glycosylation, however the paper is well described and the results and LC-MS methodology is informative. Authors approach to study the MS fragments and identifying all the passible N-glycan structures and also confirmed that enhanced fucosylation. However, this work is the in first attempt to show the potentials of MSn analysis to find out the reduced galactosyltranferase (GaIT) . I am recommending the manuscript to be published.

This manuscript is a resubmission of an earlier submission. The following is a list of the peer review reports and author responses from that submission.

Round 1

Reviewer 1 Report

The paper does not provide novel data, rather usage of a different technique to explore a known field. 

Reviewer 2 Report

Zhipeng Su and coworkers present a manuscript where the composition of IgG glycans of healthy persons and of rhreum. arthr. Patients is compared.
The approach involved affinity isolation of IgG, enzymatic release and manual permethylation with phase partition as purification method. The “new” point was the analyses by ESI-MS with apparently manual direct injection of samples. However, ESI-MS of permethylated glycans has been amply practiced decades ago and the extensive presentation of MSn data is dispensable all the more as IgG glycan structurally are not expected to bear any surprise.

Their findings are not surprising, maybe with the exception that the degree of under-galactosylation found was surprisingly low (Fig. 4A).
Even lower is the general degree of fucosylation. Actually, this sheds doubt on the experimental validity of this work!!!  Fucosylation is usually in the range of 10-12 %

Somehow interesting are the results presented in Table 5, where clearer differences are seen (e.g. in 5F).
This in turn raises the question, why the difference in 4A is so small. How was “galactosylation” calculated?

So, the approach and results part are certainly and at best confirmatory rather than innovative in any way.

The experimental approach likewise rests on classic concepts. Apparently, the samples were introduced manually. What was the injection time, inter-sample washing etc ?

Ion-trap MS is very prone to mass-based bias, i.e. to different peak heights for equal amounts of glycans of differing mass. How was this problem handled?

My fiercest criticism is the fact that neither in the introduction nor in the discussion the huge amount of data and publications that exist on RA glycosylation and other inflammatory diseases is presented and discussed. I do not recall all the pertinent data but as far as I do, the undergal ratio of patients is rather weak. As a result of ignoring the literature, differences in the results are not properly discussed. The increase in fucosylation WOULD be interesting, if I COULD believe the values insinuated by the Fig.